# Circulating Blood Prognostic Biomarker Signatures for Hemorrhagic Cerebral Cavernous Malformations (CCMs)

**DOI:** 10.3390/ijms25094740

**Published:** 2024-04-26

**Authors:** Jacob Croft, Brian Grajeda, Luis A. Aguirre, Johnathan S. Abou-Fadel, Cameron C. Ellis, Igor Estevao, Igor C. Almeida, Jun Zhang

**Affiliations:** 1Department of Molecular and Translational Medicine, Texas Tech University Health Science Center El Paso (TTUHSCEP), El Paso, TX 79905, USAjohnathan.abou-fadel@ttuhsc.edu (J.S.A.-F.); 2Department of Biological Sciences, University of Texas at El Paso, El Paso, TX 79902, USA; bigrajeda@utep.edu (B.G.); ilestevaoda@utep.edu (I.E.);

**Keywords:** hemorrhagic stroke, cerebral cavernous malformations (CCMs), circulating blood biomarker, prognostic and predictive biomarkers

## Abstract

Cerebral cavernous malformations (CCMs) are a neurological disorder characterized by enlarged intracranial capillaries in the brain, increasing the susceptibility to hemorrhagic strokes, a major cause of death and disability worldwide. The limited treatment options for CCMs underscore the importance of prognostic biomarkers to predict the likelihood of hemorrhagic events, aiding in treatment decisions and identifying potential pharmacological targets. This study aimed to identify blood biomarkers capable of diagnosing and predicting the risk of hemorrhage in CCM1 patients, establishing an initial set of circulating biomarker signatures. By analyzing proteomic profiles from both human and mouse CCM models and conducting pathway enrichment analyses, we compared groups to identify potential blood biomarkers with statistical significance. Specific candidate biomarkers primarily associated with metabolism and blood clotting pathways were identified. These biomarkers show promise as prognostic indicators for CCM1 deficiency and the risk of hemorrhagic stroke, strongly correlating with the likelihood of hemorrhagic cerebral cavernous malformations (CCMs). This lays the groundwork for further investigation into blood biomarkers to assess the risk of hemorrhagic CCMs.

## 1. Introduction

Cerebral cavernous malformations (CCMs) are a neurological disorder that causes enlarged intracranial capillaries in the brain, leading to an increased risk of hemorrhagic stroke [1,2]. This condition is particularly prevalent in the Hispanic population, with the highest rates observed in individuals with the CCM1 gene mutation [3,4,5,6]. The common Hispanic BACA-CCM1 mutation has been traced back to several generations of descendants in the West Texas borderland area, which also has the highest percentage of Hispanics in the United States and is considered the epicenter for CCMs in North America by stroke specialists. Unfortunately, the majority of individuals with the CCM gene mutation are mainly asymptomatic, but when symptoms do occur, the disease has often reached the stage of focal hemorrhage, leading to significant morbidity [7,8,9,10,11].

Currently, the treatment options for CCMs are limited to neurosurgical removal for most lesions or gamma knife radiosurgery for deep-stem CCM lesions [12,13,14,15]. These treatments are invasive and can have severe consequences for patients, highlighting the need for prognostic biomarkers to predict the risk of hemorrhagic events to inform treatment decisions better and identify future pharmacological targets. Recent research has identified several etiological serum and blood-circulating biomarkers in a selected cohort of homogeneous CCM patients sharing the BACA-CCM1 mutation [7]. Based on the previously identified biomarkers, the authors have developed the first panel of candidate biomarkers to predict the risk of hemorrhagic CCMs in the largest recorded sample size from local Hispanic CCM patients. This biomarker panel has the potential to significantly improve patient outcomes and reduce the morbidity associated with this debilitating condition.

Biomarkers are essential in predicting, screening, diagnosing, forecasting, or stratifying risk for disease outcomes. They can be cellular, histological, molecular, physiological, or radiographic characteristics and can be used alone or as a panel with multiple targets (FDA-NIH BEST resources). Blood biomarkers, especially prognostic biomarkers, are advantageous due to their low cost, feasibility, and acceptability for diagnostic and prognostic applications. They have been long sought after as diagnostic and prognostic tools for various disorders, mainly ischemic strokes [16,17,18,19,20]. Thus far, the progress in developing biomarker-driven prediction models for hemorrhagic CCMs has been restricted. A handful of initial investigations have been carried out in human and animal models, yet they have faced considerable constraints [7,21,22,23,24].

In this research, we have discovered the inaugural set of biomarker signatures closely linked to hemorrhagic risk. This innovative biomarker panel lays the groundwork for the further assessment of potential blood biomarkers in determining the risk of hemorrhagic CCMs. This could pave the way for a reformed approach incorporating updated clinical definitions and substantially enhance the management of hemorrhagic strokes.

## 2. Results

### 2.1. Differentially Expressed Serum Proteins Were Identified through Proteomics in Two Species

Using the proteomics approach, we identified 69 confirmed serum proteins from 189 peptide entries in humans and 105 confirmed serum proteins from 168 peptide entries in mice. By comparing the distribution of expressed protein profiles between CCM1-deficient subjects and normal controls across species, we found that there were more down-regulated DEPs than up-regulated DEPs in both humans (Figure 1A) and mice (Figure 1B). Of these, 35 genes were uniquely expressed and shared between the two species (Figure 1C). Comparative heatmaps visualize the differential expression of serum proteins (DEPs) between CCM1-deficient subjects and normal controls in both species (Figure 1D). This suggests that the DEPs shared by both species could serve as potential blood-circulating biomarkers for hemorrhagic CCMs. We observed a congruous trend of DEPs in both species, which was further highlighted in the volcano plots in both humans (Figure 1E) and mice (Figure 1F). The data presented showcase the involvement of DEPs in several biological processes, making them an excellent foundation for the following gene pathway and enrichment analysis.

### 2.2. CCM1-Associated Signaling Pathways Were Identified through Pathway Analyses among DEPs in Two Species

Gene pathway and enrichment analysis is a widely used method in omics to identify overrepresented gene sets in a specific subset of genes or proteins, such as pathways, gene ontology terms, or disease-associated genes. This approach provides valuable insights into the underlying biological processes of a disease phenotype. In our study, we conducted gene ontology (GO), disease ontology (DO), and KEGG pathway enrichment analyses to comparatively examine the functional characteristics and biological pathways of the differentially expressed proteins (DEPs) identified in our study.

To determine the statistical significance of the biological functions within the group of DEPs identified in our study, we first conducted group gene ontology (GO) and pathway analysis with systematic functional annotation [25,26,27]. Given the significant number of DEPs identified in both species, we used these DEPs for initial biological profiling through functional interaction network and pathway analyses. GO enrichment analysis was performed based on the described method [28]. Enriched GO pathway dot plots offer a comparative visualization of functional enrichment outcomes in both human (Figure 2A, Appendix A) and mouse (Figure 2B, Appendix A) datasets, enabling us to discern patterns in extensive biological data and assess the differentially expressed proteins (DEPs) affected by CCM1 deficiency.

The GO enrichment pathway analysis showed that human and mouse CCM1-deficient subjects shared metabolic processes and pathways compared to the respective control groups (Figure 2A,B). In contrast, the GO-based over-enrichment analysis identified shared coagulation and complement pathways in CCM1-deficient subjects relative to the control groups in both humans (Table 1A, Appendix A) and mice (Table 1B, Appendix A). These differences could be attributed to each method’s distinct bioinformatics approaches and databases.

### 2.3. Common Shared CCM1-Associated Signaling Pathways Were Identified in Two Species

We subsequently performed a disease ontology (DOSE) pathway analysis to corroborate the GO enrichment findings. This robust method facilitates the establishment of connections between enriched signaling pathways derived from differentially expressed genes and proteins (DEGs/DEPs) and the clinical phenotype (hemorrhagic CCMs), allowing for the identification of potential pathways linked to the progression of the pathology [27,29,30].

Notably, the DOSE enrichment pathway analysis revealed the sharing of both metabolic processes and pathways and coagulation and complement pathways in individuals with CCM1 deficiency compared to the control group in both humans and mice (as shown in Figure 2E,F). However, DOSE-based over-enrichment analysis only identified coagulation and complement pathways (as demonstrated in Table 2A,B), possibly due to the abovementioned reasons. These results suggest that the differentially expressed proteins associated with metabolic processes/pathways and coagulation and complement pathways could be crucial in developing CCM1 deficiency in humans and mice. These findings provide further evidence supporting the involvement of proteins from these two significant pathways in CCM1 deficiency and emphasize their potential as biomarkers for this disease.

In addition to GO and DOSE enrichment analyses, we also conducted an enriched KEGG analysis. This revealed that both human (Figure 2E, Appendix A) and mouse (Figure 2F, Appendix A) CCM1-deficient subjects exhibited alterations in coagulation and complement pathways compared to the control group, which is further supported by KEGG-based GSEA (Table 3A,B). Notably, the top three pathways identified in the enriched KEGG analysis, including complement and coagulation, ECM-receptor interaction, and focal adhesion pathways, are all closely related to hemorrhagic events. These findings offer valuable insights into the biological processes underlying CCM1 deficiency and may contribute to developing new therapeutic approaches for the disease.

### 2.4. Candidate Serum Circulating Biomarkers Were Identified through Comparative Analysis among Common Shared CCM1-Associated Signaling Pathways in Two Species

Finally, we conducted a comprehensive comparative analysis among three separate enrichment approaches (GO/DOSE/KEGG) to identify pathways affected by CCM1 deficiency in comparison to the control group across both species. Our comparative pathway dot plots reveal that both metabolic processes/pathways and coagulation and complement pathways are consistently present in the sera of subjects with CCM1 deficiency relative to the control group in both humans and mice (Figure 2G). This observation is further substantiated by the GSEA results derived from the three distinct enrichment libraries (Table 3A,B). Our findings indicate that the identified serum DEPs are mainly linked to two primary pathways that could act as potential circulating blood biomarkers for the disease: metabolic processes/pathways and coagulation and complement pathways. Recent research in our lab has shown that CCM1 loss-of-function (LOF) results in disrupted metabolic processes/pathways, which aligns with prior studies highlighting the association between metabolic disturbances, oxidative stress, and intracellular reactive oxygen species (ROS) with CCM1 deficiency [31,32,33,34]. Additionally, we identified serum DEPs associated with coagulation and complement pathways, including complement C3, fibrinogen, vitronectin, collagen, fibronectin, and laminin. These findings align with recent studies. In total, we discovered 21 potential blood biomarkers among the 71 detected DEPs. This study identified 21 potential blood biomarkers, primarily within metabolic processes/pathways and the complement and coagulation cascade pathway, that could serve as etiological and prognostic blood biomarkers, respectively. Table 4. These biomarkers were found in BACA CCM1 hemizygous mutation patients and Ccm1 hemizygous mutant mice using three distinct pathway enrichment methods with separate library datasets. The consistent results across approaches reinforce the findings and emphasize the potential of these biomarkers as diagnostic tools for CCM1 deficiency.

## 3. Discussion

The objective of this research is to systematically detect prospective blood prognostic biomarkers in a homogeneous group of Hispanic fCCM patients and Ccm mutant mice, establishing a solid basis for our ongoing biomarker project. Our ongoing project aims to assess the potential of these biomarkers to predict early hemorrhagic events and recognize a critical time window for patients to receive preventive and therapeutic treatment using deep learning algorithms combined with our candidate serum biomarkers. The proposed experiments offer valuable insights into the development and prognosis of hemorrhagic CCMs and provide information about environmental exposures, effect modifiers, or risk factors linked to their progression. Gene set enrichment analysis (GSEA) was used to identify signaling pathways enriched due to CCM1 deficiency. The top two pathways identified were metabolic processes/pathways and coagulation and complement pathways, which are relevant to hemorrhagic events. Coagulation signaling is linked to hemorrhagic stroke risk and outcomes [35,36,37,38,39], leading to proposals for coagulation-targeted therapies and circulating prognostic biomarkers [36,40]. Similarly, the complement cascade has also been associated with hemorrhagic stroke. However, it is still under debate whether complement factors can be used as prognostic tools for pre-hemorrhagic progression or as biomarkers for recovery after hemorrhagic events [35,41,42,43,44].

Among identified candidate prognostic biomarkers, plasma kallikrein (PKa) is involved in blood coagulation, fibrinolysis, hemostasis, and inflammatory response [45,46,47,48]. PKa deficiency due to KLKB1 mutations leads to vascular bleeding and has been implicated in hereditary angioedema and hemorrhagic stroke [47,49,50,51,52,53,54,55]. Serpins, a superfamily of serine protease inhibitors, play critical roles in vascular angiogenesis and have been implicated in retinal vascular leakage and hemorrhagic stroke [56,57,58,59,60,61,62,63]. Peptidoglycan recognition protein 2 (PGLYRP2) is involved in immunomodulation and innate immunity, while the adenomatous polyposis coli (Apc) gene is crucial in development, negatively regulates Wnt signaling, and may be involved in angiogenesis [64,65,66,67,68,69]. Retinol binding protein 4 (RBP4) is linked to the severity of cardiovascular disorders, and complement factors are known to be associated with hemorrhagic stroke [35,41,42,43,44,70,71,72,73]. These biomarkers may help in understanding and treating various vascular conditions. Several limitations to this study should be acknowledged. Firstly, the analysis did not include non-hemorrhagic CCMs (NHCs) due to the experimental design of the omics. Secondly, while this study has the largest sample size with both human and mouse data, our power analysis indicates that with the current sample size, we may have missed some potential targeted proteins, leading to type 2 errors. To address this, we plan to examine 19 subjects and 38 controls with a 1:2 ratio to achieve 80% power to detect significant differences with a 0.05 significance level. Thirdly, while we recognize that our comparison involved hemorrhagic CCM patients with a shared mutated CCM1 gene and healthy subjects with a wild-type CCM1 gene background, this study served as our initial exploration of etiological biomarkers for hemorrhagic events. However, recent discoveries regarding the genetic heterogeneity of both familial and sporadic CCMs, as well as the various potential genetic modifiers and environmental triggers for hemorrhagic events in CCM patients, suggest the need to enhance our current findings. This could entail further comparative analysis among candidate biomarkers identified from familial and sporadic CCM studies. Alternatively, conducting a longitudinal study to compare healthy, non-symptomatic, or pre- and post-hemorrhagic subjects within familial CCM cohorts, all carrying specific CCM gene mutations, with genomic, proteomic, and systems biological analysis approaches would be beneficial. We believe that this integrative approach can yield more precise and sensitive biomarkers for diagnostic, prognostic, and predictive purposes, potentially enabling the prediction and prevention of the most severe and life-altering event, the hemorrhagic stroke. Henceforth, regarding the future trajectory of the project, our strategy to tackle the challenges and execute the proposed potential solutions mentioned earlier entails conducting analyses with a significantly larger sample size of subjects and diverse control groups, leveraging multi-center collaborative efforts. This strategy aims to streamline the identification of biomarkers capable of detecting subtle differences between non-hemorrhagic and hemorrhagic subjects. Being part of the international pooled CCM data consortium, we hold a strong belief in our ability to achieve this objective in the foreseeable future.

## 4. Materials and Methods

The objective of this study was to identify a set of blood-based biomarkers that can predict the prognosis of different stages of hemorrhagic strokes. To accomplish this, we began an exploratory project with the hypothesis that prognostic blood biomarkers for hemorrhagic risk, discovered within a genetically well-defined familial CCM cohort, can be expanded and applied to sporadic CCM cases and ultimately to a broader range of hemorrhagic strokes. Moreover, ongoing debates and unresolved issues persist concerning the underlying causes of both familial and sporadic CCM forms, especially whether they originate from familial CCM mutations or other unknown genes [2,11,74,75,76,77]. The future validation of potential biomarkers stemming from this research will help address this query.

Since this is a comprehensive investigation of CCM1 mutation effects and serves as a basis for further large-scale studies, the participants were chosen based on their homogenous genetic predisposition and symptomatic presentation to establish the differential protein expression patterns of CCM1 mutations compared to those of age-/gender-matched control groups. We are confident in analyzing the experimental outcomes with a cost-effective model for protein expression data to minimize type 1 errors (type 1 errors = 0.05) at the cost of sacrificing for type 2 errors (missing some potential targeted proteins), as detailed in the statistical analysis section [78]. In this study, we carried out a proteomics analysis involving a cohort of human patients with familial cerebral cavernous malformations (fCCMs) resulting from a uniform BACA CCM1 hemizygous mutation alongside their age- and gender-matched healthy controls (*n* = 14). In addition, we examined Ccm1 mutant mice along with their wild-type (WT) counterparts (*n* = 6) to strengthen our analytical capabilities. While this omics research substantially deviates from traditional epidemiological analysis, it is important to note that this investigation is conducted within the context of a clinical trial focused on biomarker identification and validation. As a result, we have rigorously adhered to the STROBE (Strengthening the Reporting of Observational Studies in Epidemiology) guidelines [79] throughout the process of preparing this manuscript.

### 4.1. fCCM Patient Cohort Recruitment Procedure

The criteria for the International Classification of Diseases (ICD) codes 9/10 must be met for a patient’s medical history to be considered for the study on CCMs (categorized as 228.00, 228.02, 228.09/Q28.3, D18.00, D18.02). Neurovascular disorders (747.81/Q28.2, Q04.9, G93.9) and codes related to hemorrhagic stroke and epilepsy (430, 431, 432.1, 432.9, and 345.00, 345.01/I60.9, I61.9, I62.00, I62.9 and G40.A01, G40.A09, G40.A11, G40A.19) may also be used as supplementary criteria. The authorized IRB protocol permits the enrollment and consent of participants ranging from 8 to 89 years old. Considering the substantial Hispanic population in the vicinity, the study primarily focuses on including minority individuals, although it does not impose any restrictions, as detailed in. Individuals with CCMs can be classified into two distinct groups: non-hemorrhagic CCMs (NHCs) and hemorrhagic CCMs (HCs). Only hemorrhagic CCMs (HCs) and healthy controls (Ctrls) were utilized for this comparative proteomic study.

### 4.2. Data Collection

Data for the study were collected through structured interviews conducted in person by clinical co-investigators and stored on secure PCs. The correlation between blood biomarker levels and disease severity in fCCM patients was analyzed using statistical methods. Odds ratios and 95% confidence intervals were calculated using stratified data analysis and logistic regression to determine the relationship between the blood levels of biomarkers and the risk of hemorrhagic stroke in fCCM patients. Comparisons were made between symptomatic fCCM carriers (HCs) and healthy controls to determine the correlation between blood levels of biomarkers and the odds of hemorrhagic stroke. All statistical tests were two-tailed with a significance level of *p* < 0.05.

Given that all individuals in this cohort share the same Mendelian rare causal variant, specifically a single Hispanic CCM1 mutation responsible for their CCM disorder, this homogeneous group with a rare genetic mutation demonstrates a limited sample size yet a significant effect size due to the mutant allele’s substantial influence on disease progression. This strategy of utilizing a small sample size with a larger impact has a proven track record of success in genetic studies, such as linkage analysis for identifying disease-causing alleles or genes. Thus, we postulated that this rare genetic mutation exerts considerable influence with ample power, facilitating the discovery of disease-associated biomarkers at both transcriptional and translational levels. Based on this reasoning, to further validate this experiment, both human and mouse species bearing the same mutation acted as replicas of each other, accompanied by gender- and age-matched healthy controls for normalization. Subsequently, the datasets were merged for comparative analysis, facilitating further reassessment and validation of the identified candidate biomarker pool. This iterative process contributed to the finalization of candidate biomarkers for evaluating disease risk during pathway enrichment analysis.

### 4.3. Biomarker Data Collection from Proteomics

We conducted a comprehensive and impartial search to discover new biomarkers through high-throughput omics methods using human serum samples from fCCM patients and healthy matched controls. Our optimized procedures include: *(3-1) Abundant serum protein depletion*: We processed ten serum samples from hemorrhagic fCCM patients with BACA-CCM1 mutations and matched controls for proteomic analysis [80,81,82,83]. The total protein concentration was determined using a bicinchoninic acid (BCA) assay (Bio-Rad, Hercules, CA, USA). Then, using High Select Mini Spin Columns (Thermo Scientific), we removed highly abundant proteins, such as albumin, immunoglobulins, fibrinogen, and transferrin, to detect low-abundance biomarkers. The protein concentration was re-measured using the BCA assay. *(3-2) Protein Enrichment*: The serum samples were processed with a ProteoMiner kit (Bio-Rad), and the protein concentration was determined using the BCA and fluorescence assays. *(3-3) Trypsin Digestion and iTRAQ Labeling*: The serum proteins were reduced, alkylated, desalted, buffer changed, and then digested with trypsin and labeled with iTRAQ reagents [80,81,82,83]. The labeled peptides were subjected to a Phoenix Peptide Cleanup Kit to remove excess labeling and salt and were lyophilized and stored at −80 °C. *(3-4) Raw Proteomic Data Acquisition*: All proteomic data were acquired through tandem mass spectrometry (MS/MS) analysis, including NanoLC-MS/MS analysis, using a Q Exactive Plus Hybrid Quadrupole-Orbitrap Mass Spectrometer (Thermo Scientific, Waltham, MA, USA) [80,81,82,83].

After implementing these refined proteomic procedures, our objective was to discover novel biomarkers in fCCM patients relative to healthy matched controls using high-throughput omics techniques, employing human serum samples as previously outlined [80,81,82,83].

### 4.4. Biomarker Identification through Pathway Analyses with Bioinformatics Tools

The Bioconductor R package was employed to conduct biological process and functional interaction network analyses [84,85,86]. The groupGO function in clusterProfiler of Bioconductor was used to determine the functional profile of the pathway components by grouping genes based on their gene ontology [87,88,89,90]. A vector of UniProt Accession numbers was provided for the gene argument, and the Bioconductor Genome-Wide Annotation for Mouse was used as the database. The keytype argument was set to “UNIPROT”, converting the UniProt Accession numbers into Entrez Gene IDs, and the level was set to GO levels 2. This process was repeated for all three sub-ontologies and followed by hierarchical clustering analysis. Complementing this process, the enrichGO function in clusterProfiler of the Bioconductor further explored the grouping of the gene ontology, creating pathways visualizing functional profiles of where these proteins were located in the previously mentioned sub-ontology levels. GO, KEGG, and DOSE pathways, modules, and pathway enrichment analysis were performed using Bioconductor and its associated functions [29,87,91,92,93,94,95,96]. The enrichDO function within the clusterProfiler package of Bioconductor was employed as a systematic approach to analyze disease ontology based on shared genes mapped from Entrez Gene IDs [29]. Subsequently, pathway enrichments were conducted by analyzing Entrez Gene IDs to discern shared molecular functions and relationships across subjects.

### 4.5. Statistical Analysis

Significance was meticulously adjusted for multiple comparisons, and differences between groups were rigorously compared using the Student’s *t*-test or paired samples t-test. A *p*-value less than 0.05 was considered statistically significant. Our analysis, conducted with the utmost care, indicated that our proposed sample size would be suitable for exploring protein expression differences between CCM1 deficient and normal groups (HCs and age-/gender-matched Ctrls). Results were visualized using ggplot2 with log2fc for fold changes, providing a clear and comprehensive representation of the data [97].

A power analysis was employed to assess the ability of the sample sizes to yield significant findings. By setting a significance level of 0.05 to reduce type 1 errors, the power level dropped below 80%, making the study more prone to type 2 errors or false negatives. Only participants with a homogenous genetic background were recruited to counteract this, ensuring a population with a known inheritance history. The selection of subjects was further refined by considering phenotypical manifestations in a clinical setting, which guaranteed a sample with actively expressed proteins relevant to the condition. Consequently, instead of relying on a power analysis-based population, a cost-effective model was adopted, resulting in statistically significant findings contributing to the field and the potential identification of biomarkers.

Additionally, a Benjamini–Hochberg procedure was utilized as the FDR method throughout the enrichment process. We then used only *p*-adjusted values to prevent any false positive findings of the overrepresentation analysis. While this makes us more prone to type-2 false negatives, we utilized the limitations to create only limited, accurate findings to focus on only the highest related pathways in this analysis. We only accepted statistically significant findings when the *p*-adjusted value was less than 0.05.

## 5. Conclusions

The aim of this study was to identify an initial set of circulating blood biomarker signatures for diagnosing and predicting hemorrhagic risk in CCM1 patients. Both human and mouse subjects displayed varying expression of serum proteins (DEPs). We employed combined enrichment analysis methods, encompassing gene ontology (GO), disease ontology (DOSE), and KEGG pathway analysis, to explore the functional characteristics and biological pathways linked to CCM1 deficiency-related DEPs. Through bioinformatic analysis of blood proteomic data from Hispanic individuals and CCM1-deficient mice, we identified a specific panel of blood biomarkers primarily associated with metabolism and blood clotting pathways. These biomarkers hold potential as prognostic indicators for CCM1 deficiency and the risk of hemorrhagic stroke. Moreover, they exhibit a significant correlation with the likelihood of hemorrhagic cerebral cavernous malformations (CCMs). This underscores the need for further evaluation and validation of these candidate blood biomarkers in a larger CCM cohort with a more heterogeneous genetic background to reassess their clinical potential for assessing hemorrhagic CCM risk.

## Figures and Tables

**Figure 1 ijms-25-04740-f001:**
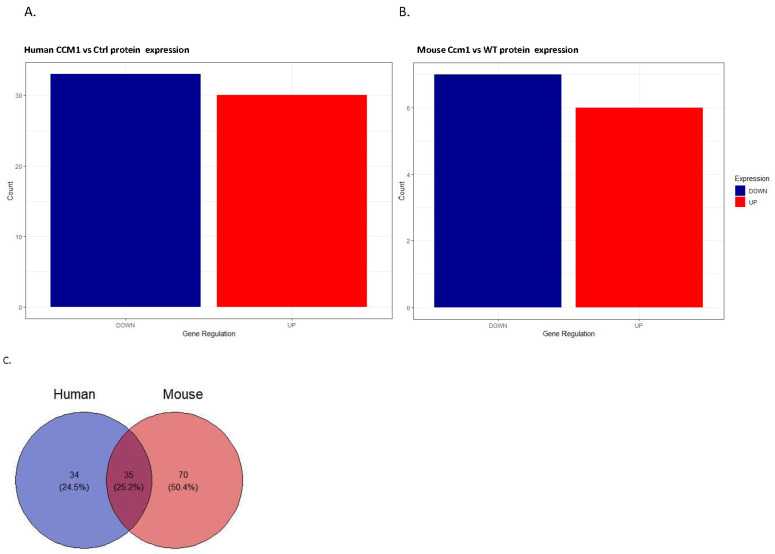
Differentially expressed serum proteins (DEPs) were identified in CCM1-deficient subjects and normal controls across species using high-throughput comparative proteomics. The statistical significance of differentially expressed serum proteins (DEPs) was determined by analyzing the distribution of expressed protein profiles. (**A**,**B**) Highly expressed serum proteins were distributed across species in CCM1-deficient subjects and normal controls. The distribution of DEPs was overlaid by comparing CCM1-deficient subjects to normal controls in both humans (**A**) and mice (**B**). Up- (red bar) and down- (blue bar) regulation were determined by fold change values. (**C**,**D**) Unique patterns in the DEPs were identified by comparing CCM1-deficient individuals and normal controls in two species. A Venn diagram displays unique and co-expressed proteins between humans and mice (**C**). Comparative heatmaps illustrate protein expression changes between CCM1-deficient subjects and controls in both species (**D**). The heatmaps display protein expression values for each related protein on the *Y*-axis. (**E**,**F**) Volcano plots display DEPs between CCM1-deficient subjects and healthy controls. This visualization highlights proteins in both control and confirmed CCM-deficient subjects, demonstrating how the differentiated protein expression varies compared to controls. A *p*-value, derived from a Student’s *t*-test, is shown in a −log10 function relative to the log2 fold change. This approach helps regulate infinite fold values for unique values, preventing graph distortion. The data points within the plot represent fold changes in protein expression for CCM1-deficient subjects compared to the control group in both humans (**E**) and mice (**F**). The values of up-and down-regulated proteins from figures (**E**,**F**) were refined using a −log10 *p*-value threshold to identify more significant results. For specific gene/protein names corresponding to the dots in the plot, please refer to Appendix A.

**Figure 2 ijms-25-04740-f002:**
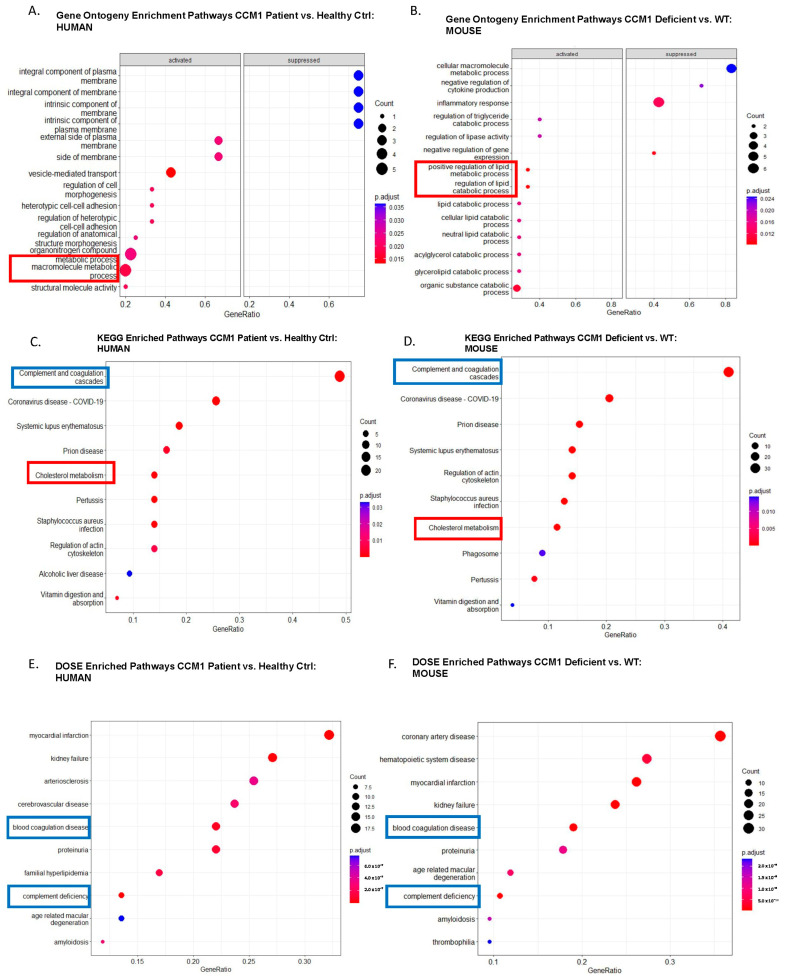
Comparative proteomics identifies critical pathways and their associated biomarkers for CCM1 deficiency. Comparative proteomics research has revealed crucial pathways and biomarkers related to CCM1 deficiency. By detecting differentially expressed serum proteins (DEPs) in CCM1-deficient and normal subjects, the study successfully uncovered various pathways implicated in CCM1 deficiency. This was achieved through gene set enrichment analysis (GSEA) of the identified DEPs in human and mouse subjects. (**A**,**B**) Gene ontology (GO) groupings for functional interaction networks and pathway analyses were conducted to assess the statistical significance of a biological process among the identified DEPs. This was done to evaluate the enrichment of differentially expressed serum proteins (DEPs) between CCM1-deficient and normal conditions in both humans (**A**) and mice (**B**). Two common pathways between (**A**) and (**B**) were identified: the cellular metabolic pathway and the coagulation and complement pathway. (**C**,**D**) A comparative proteomics study was carried out between CCM1-deficient and normal conditions, employing the Kyto Encyclopedia of Genes and Genomes (KEGG) pathways to enrich differentially expressed serum proteins (DEPs) in humans (**C**) and mice (**D**). The pathways emphasized by colored frames represent the top-selected pathways. Two common pathways between (**A**) and (**B**) were identified: the cellular metabolic pathway and the coagulation and complement pathway. (**E**,**F**) The disease ontogeny (DOSE) pathways were employed to enrich differentially expressed serum proteins (DEPs) between CCM1-deficient and normal conditions in humans (**E**) and mice (**F**). Two common pathways between (**A**) and (**B**) were identified: the cellular metabolic pathway and the coagulation and complement pathway. (**G**) Applying three distinct enrichment libraries showcases pathways impacted by the CCM1 deficiency compared to the control for both species studied. From left to right, we observe enriched gene ontology (GO), Kyoto Encyclopedia of Genes and Genomes (KEGG), and Disease Ontology Semantic and Enrichment (DOSE) analyses, revealing affected pathways in metabolism (red outline) and complement and coagulation cascade pathways (blue outline). The highlighted pathways across two different species hold the potential for discovering etiological and prognostic biomarkers for future validation analysis.

**Table 1 ijms-25-04740-t001:** Gene set enrichment analysis (GSEA) for GO. The gene set enrichment analysis (GSEA) was performed using the gene ontology (GO) of CCM1-deficient subjects compared to that of the control in both humans (**A**) and mice (**B**). Significant DEP enrichment was then compared to a human database from BiocManager (Org.Hs.eg.db) (**A**) and a mouse database from BiocManager (Org.Mm.eg.db) (**B**). The DEPs associated with pathways that showed significant differences in GO enrichment analysis are listed in the table. The pathways highlighted in blue represent the complement and coagulation cascade, with associated DEPs potentially serving as prognostic biomarkers.

**(A)**		
**Ontology**	**Description**	**Genes Involved**
BP	negative regulation of multicellular organismal process	FN1 APOD PGLYRP2 RBP4 KNG1 APOA2 APCS VTN APOE KLKB1 SERPINF2 APOC3
BP	vesicle-mediated transport	APOA2 VTN APOE PCLO APOC3
BP	regulation of vesicle-mediated transport	APOA2 VTN APOE PCLO APOC3
MF	phospholipid binding	APOE PCLO APOC3
BP	regulation of wound healing	KNG1 APCS VTN APOE KLKB1 SERPINF2
BP	negative regulation of cellular process	APOD PGLYRP2 RBP4 KNG1 C5 APOA2 APCS APOE APOC3
**(B)**		
**Ontology**	**Description**	**Genes Involved**
BP	Cytokine Production	Thbs1 Serpinf1 B2m
BP	Regulation of Cytokine Production	Thbs1 Serpinf1 B2m
BP	Behavior	Thbs1 Serpinf1 B2m
BP	Negative Regulation of Gene Expression	Thbs1 Pglyrp2 Serpinf1
BP	Immune Response	Pglyrp2 Igkv1-135 Kng2 Igkv4-70 APCS Ighv6-6 B2m
CC	Extracellular region	C8b Pf4 Klkb1 Gpx3 Qsox1 F12 RBP4 Gsn Cd5l F5 Lcat Agt Thbs1 Pglyrp2 Igkv1-135 Serpinf1 Kng2 F13a1 Igkv4-70 APCS Ighv6-6 B2m

**Table 2 ijms-25-04740-t002:** Gene set enrichment analysis (GSEA) using DOSE. The GSEA was performed based on the disease ontology (DOSE) of CCM1-deficient subjects compared to that of the control in both humans (**A**) and mice (**B**). Significant DEP enrichment was then compared to a human database from BiocManager (Org.Hs.eg.db) (**A**) and a mouse database from BiocManager (Org.Mm.eg.db) (**B**). The DEPs associated with pathways that showed significant differences in GO enrichment analysis are listed in the table. The pathways highlighted in blue represent the complement and coagulation cascade, with associated DEPs potentially serving as prognostic biomarkers.

**(A)**		
**DO:ID**	**Description**	**Gene Symbol**
DOID:6262	Complement Deficiency	APOA1, C3, C4B, C5, C6, C7, C8A, C8B, CFH, CFHR1
DOID:1247	Blood Coagulation Disease	C3, C4B, C5, CFH, CPB2, F11, F13B, FGA, PROS1, SELL, SERPINA10, SERPIND1, SERPINF2, VWF
DOID:5844	Myocardial Infarction	APOA1, APOA4, APOE, C3, C4B, C7, C8A, C8B, CFH, CPB2, F11, F13B, FGA, GSN, ITIH4, LPA, SERPIND1, VWF
DOID:10871	Age Related Macular Degeneration	APOE, C3, C5, CFD, CFHR1, SELL, SERPINF1, TF
DO: 576	Proteinuria	APOA1, APOA4, APOE, C3, C4B, C5,C6, CFH, CPB2, HPX, LPA, VTN
DO:1074	Kidney Failure	AMBP, APOA1, APOA4, APOE, C3, C6, CFH, CPB2, FGA, PROS1, S100A8, SELL, SERPINF1, TFRC, VWF
**(B)**		
**DO: ID**	**Description**	**Gene Symbol**
DOID:3393	Coronary Artery Disease	APOA1, APOA2, APOA4, APOC3, APOC4, APOE, APOM, C3, C7, C8A, C8B, CFH, CPB2, F11, F13B, F5, FGA, GSN, ITIH3, ITH4, LCAT, LPA, LUM, PON3, SELL, SERPIND1, SERPINF2, SERPING1, VTN, VWF
DOID:626	Complement Deficiency	APOA1, C3,C5, C6, C7, C8A, C8B, CFH, CFHR1
DOID:1074	Kidney Failure	AMBP, APOA1, APOA4, APOC3, APOE, B2M, C3, C6, CFH, CPB2, F5, FGA, LCAT, PROS1, PTGDS, S100A8, SELL, SERPINF1, TFRC, VWF
DOID:5844	Myocardial Infarction	APOA1, APOA4, APOC3, APOE, C3, C7, C8A, C8B, CFH, CPB2, F11, F13B, F5, FGA, GSN, ITIH3, ITIH4, LCAT, LPA, SERPIND1, SERPING1, VWF
DOID:1247	Blood Coagulation Disease	C3,C5, CFH, CP, CPB2, F11, F13B, F5, FGA, KLKB1, PROS1, SERPINA10, SERPIND1, SERPINF2, VWF
DOID:74	Hematopoietic System Disease	APOE, C3,C5,CA1,CP, CPB2, F11, F13B, F5, FGA, KLKB1, MASP2, SELL, SERPINA10, SERPIND1, SERPINF1, SERPINF2, TF, TFRC, VWF

**Table 3 ijms-25-04740-t003:** Gene set enrichment analysis (GSEA) based on KEGG. The GSEA was performed based the Kyoto Encyclopedia of Genes and Genomes (KEGG) pathways of CCM1-deficient subjects compared to those of the control in both humans (**A**) and mice (**B**). Significant DEP enrichment was then compared to a human database from BiocManager (Org.Hs.eg.db) (**A**) and a mouse database from BiocManager (Org.Mm.eg.db) (**B**). Blue-highlighted pathways represent the complement and coagulation cascade, with associated DEPs potentially serving as prognostic biomarkers. Furthermore, red-highlighted pathways pertain to the metabolic pathway, where related DEPs are intended for evaluation as potential etiological biomarkers.

**(A)**		
**ID**	**Description**	**Gene Symbol**
hsa04610	Complement and coagulation cascades	A2M, C1QA, C1QC, C3, C4BP, C6, C8A, F11, F13B, FGA, MASP2, SERPIND1, SERPINF2, VTN VWF
hsa05171	Coronavirus disease—COVID-19	C1QA, C1QC, C3, C6, C8A, F13B, FGA, MASP2, VWF
hsa05322	Systemic lupus erythematosus	C1QA, C1QC, C3, C6, C8A, H2AJ
hsa05150	Staphylococcus aureus infection	C1QA, C1QC, C3, KRT9, MASP2
hsa05133	Pertussis	C1QA, C1QC, C3, C4BPB
hsa04613	Neutrophil extracellular trap formation	C3, FGA, H2AJ, VWF
hsa04512	ECM-receptor interaction	CD44, VTN, VWF
hsa05142	Chagas disease	C1QA, C1QC, C3
(**B**)		
**ID**	**Description**	**Gene Symbol**
hsa04610	Complement and coagulation cascades	C1QA, C1QB, C1QC, C3, C4BPB, C5, C6, C7, C8A, C8B, CFD, CFH, CFHR1, CPB2, F11, F13B, F5, FGA, KLKB1, MASP2, PROS1, SERPIND1, SERPINF2, SERPING1, VTN, VWF
hsa05171	Coronavirus disease—COVID-19	C1QA, C1QB, C1QX, C3, C5, C6, C7, C8A, C8B, CFD, F13B, FGA, IKBKG, MASP2,VWF
hsa05150	Staphylococcus aureus infection	C1QA, C1QB, C1QC, C3, C5, CFD,CFH, KRT9, MASP2
hsa04979	Cholesterol metabolism	APOA1, APOA2, APOA4, APOC3, APOE, LCAT, LPA
hsa05322	Systemic lupus erythematosus	C1QA, C1QB, C1QC, C3, C5, C6, C7, C8A, C8B, H2AJ
hsa05133	Pertussis	C1QA, C1QB, C1QC, C3, C4BPB, C5, SERPING1

**Table 4 ijms-25-04740-t004:** The combined gene set enrichments from three distinct enrichment libraries showcase pathways impacted by the CCM1 deficiency compared to the control for both species. Blue-highlighted pathways represent the complement and coagulation cascade, with associated DEPs potentially serving as prognostic biomarkers. Additionally, red-highlighted pathways pertain to the metabolic pathway, where related DEPs are intended for evaluation as potential etiological biomarkers.

Gene Symbol	Gene Name	Pathway
APCS	Amyloid P Component	Complement and coagulation cascades
SERPINF1	Serpin Family F Member 1	Complement and coagulation cascades
SERPINF2	Serpin Family F Member 2	Complement and coagulation cascades
THBS1	Thrombospondin 1	Complement and coagulation cascades
VTN	Vitronectin	Complement and coagulation cascades
FN1	Fibronectin 1	Complement and coagulation cascades
F12	Coagulation Factor XII	Complement and coagulation cascades
GSN	Gelsolin	Complement and coagulation cascades
KLKB1	Kallikrein B1	Complement and coagulation cascades
KNG1	Kininogen 1	Complement and coagulation cascades
KNG2	Kininogen 2	Complement and coagulation cascades
LCAT	Lecithin-Cholesterol Acyltransferase	Cholesterol metabolism
APOA2	Apolipoprotein A2	Lipoprotein metabolism
APOC3	Apolipoprotein C3	Lipoprotein metabolism
APOD	Apolipoprotein D	Lipoprotein metabolism
APOE	Apolipoprotein E	Lipoprotein metabolism
PCLO	Piccolo Presynaptic Cytomatrix Protein	Cytoskeletal matrix
QSOX1	Quiescin Sulfhydryl Oxidase 1	Extracellular matrix
B2M	β2-Microglobulin	Immunopathways
RBP4	Retinol Binding Protein 4	Membrane transporter
PGLYRP2	Peptidoglycan Recognition Protein 2	*N*-acetylmuramoyl-l-alanine amidase

## Data Availability

Appendix A in the online version of the journal provide some essential analytical data, while all data submitted were compliant with Institutional or Ethical Review Board requirements and applicable government regulations. Any additional requests for specific portions of the original datasets will be fulfilled by the corresponding author upon request, in accordance with their institutional intellectual property policy.

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
