# Peer review of "Circulating Blood Prognostic Biomarker Signatures for Hemorrhagic Cerebral Cavernous Malformations (CCMs)"

_ijms, 2024, doi:10.3390/ijms25094740_

Round 1
Reviewer 1 Report
Comments and Suggestions for Authors
The manuscript “Circulating blood prognostic biomarker signatures for hemorrhagic
cerebral cavernous malformations (CCMs)” is interesting, well-written, and easy to read. Experiments aimed at identifying biomarkers of the risk of hemorrhagic stroke in CCM using human and animal material and several databases allowed the identification of several candidates. Biomarkers mentioned in the manuscript seem to exhibit a strong association with the hemorrhage possibility in CCM patients.
It is a pity that it cannot be said whether these biomarkers can be used only for patients with the mutation tested by the authors or for all forms of CCM. This may be a starting point for further research.
Generally, the presented manuscript can be published without additional corrections.
Author Response
Please see attached response, thank you!

Reviewer 2 Report
Comments and Suggestions for Authors
This is a seemingly very meaningful biomarker study of the risk of secondary bleeding in CCM, but the results presented by the authors do not meet their ambitious goals. On the contrary, the results seem so unprofessional and, in the end, offer no clear conclusions.
1. The elements that should be included in the Abstract are incomplete. In the abstract, the author stated the purpose, significance and analysis method in this study, but the explanation of the results and key findings in the study was lacking. Additional modifications are required.
2. The sketchy description of the details of the subjects raised confusion and doubts about the protein markers associated with bleeding risk that they analyzed. Whether all or some of these CCM patients had intracerebral hemorrhage, and whether the mice were identified and evaluated for hemorrhage prior to sampling? If there are bleeding and non-bleeding patients in the family, it is necessary to conduct a comparative analysis between them, which may be more helpful to screen and determine the factors associated with bleeding risk.
3. The bioinformation analysis in the whole article is extremely unprofessional, as the quality of these illustrations is poor, and the labeling is extremely irregular. For example, the volcano map of differentially expressed proteins is not labeled at all, giving the reader very limited information. In functional analysis fig 2A, the basis and purpose of the enrichment pathway selected by the authors were not understood.
4. The conclusions of the study were inconclusive to the end. From these analyses, the authors do not appear to have conclusively identified which marker proteins are reliable predicters of secondary bleeding risk in CCM. Functional enrichment analysis alone cannot be used as a marker for clinical prediction of secondary bleeding risk in CCM. For marker proteins in peripheral blood that may indicate the risk of CCM secondary bleeding, correlation analysis and verification should be carried out in more clinical samples, which is very important for clinical application.
Comments on the Quality of English LanguageMinor editing of English language and typesetting are required.
Author Response

(The authors gave the same response as above.)

Reviewer 3 Report
Comments and Suggestions for Authors
The manuscript "Circulating Blood Prognostic Biomarker Signatures for Hemorrhagic Cerebral Cavernous Malformations (CCMs)" focuses on identifying prognostic biomarkers for hemorrhagic CCMs through a comparative proteomic analysis between hemorrhagic CCM patients and healthy controls while also utilizing an additional mouse model system. The area of focus on biomarker identification for CCM is relevant and needed. The methodological approach using proteomics to develop blood-based biomarkers has definite utility.
In this study, the authors have identified a small set of differentially expressed proteins in human patients and a mouse model in a test vs. control setup. while the authors acknowledge the sample size limitation and the exclusion of non-hemorrhagic CCMs, which hampers the generalizability of the study to the full spectrum of CCM pathology and biomarkers. Given the small sample size, the methods section must be elaborated further. While the small sample size can be considered when using homogenous samples to find small but clear signals, the authors do not mention any details on using replicates (technical or otherwise) in the study. This is a critical part of the study design, especially given the limited number of samples. Further, since the study entails biomarker identification using a proteomic approach, additional technical details are required on the proteomic protocol used. The details related to normalization steps conducted in experiments and proteomic data analysis are critical and must be elaborated further. How and what was the cutoff for unique peptides/proteins identified using mass spec? Further steps in the statistical analysis need to be clearly defined. At a minimum, a more detailed justification for the choice of statistical tests could enhance the transparency and replicability of the findings.
Additionally, the figures should be improved, and the identified key DEPs must be labeled in the volcano plots to enhance the ease of interpretability for the readers. 1-D gene names overlap and need work. The multiple correction methods employed and cut-off used in KEGG, GO, and GSEA, as well as other analyses, need to be specified clearly in the methodology section and not just as a non-descriptive cut-off in the figures. What is the p.adjust methodology employed? FDR/ Bonferroni/others? The methodological section needs detailed attention before publication.
When doing a comparative analysis, the authors mention an overlap of proteins identified between human and mouse hemorrhagic CCM systems. This is vague, and the authors must clarify whether the abundance metrics expression changed in the same direction or was inversed between the two test-control groups. An UpSet plot or barcode plot would be helpful here. The differential expression analysis for biomarkers led to a small number of unique proteins being identified, and I suspect a more rigorous approach might not identify these as statistically significant. While ontology and enrichment can be helpful, I can only comment further on the significance of the findings if there is further methodological clarity.
The paper also does not provide details on the external validation of the identified biomarkers, which is crucial for confirming any predictive value. Given these limitations, unfortunately, I would have to reject the paper in its present state.
Author Response

(The authors gave the same response as above.)

Round 2
Reviewer 2 Report
Comments and Suggestions for Authors
Although the authors made point-to-point modifications to my comments, in my opinion, their modifications did not substantially improve the quality of the results or the accuracy of the conclusions.
1. The author did not give much thought to the second comment I mentioned. The objective of this study was to identify serum protein markers associated with the risk of bleeding from CCM. Because they wanted to accurately assess markers associated with bleeding risk, samples of CCM patients without bleeding were good controls for bleeding, not normal healthy controls. In this study, the authors discarded samples from patients who were diagnosed with CCM but did not bleed, which makes it likely that a simple comparison with a normal healthy group contains many similar biomarkers for bleeding related diseases.
2. The quality of the data in the author's revised manuscript has not been significantly improved, which greatly limits the reader's reading and understanding.
3. This sentence in the abstract is not necessary for a review of previous studies---“Recent research has identifed several serum and blood-circulating biomarkers in a selected cohort of homogeneous CCM patients and animal models. Proteomic profles from both human and mouse CCM models were examined, and pathway enrichment analyses were performed using three
approaches (GO, KEGG, and DOSE). Multiple comparisons were utilized to pinpoint potential blood biomarkers by assessing variations between groups, with statistical signifcance defned as a p-adjusted value below 0.05." The authors only need to clarify the research methods and important findings of this study.
None.
Author Response
Please see attached, thank you!

Reviewer 3 Report
Comments and Suggestions for Authors
I appreciate the efforts of the authors to address the concerns raised by me. I have gone through the revised version of the manuscript and find that the authors have duly expanded the methods section and addressed other major concerns, thereby significantly improving the manuscript. I am satisfied that these enhancements through the peer review process have sufficiently improved the manuscript and will be of particular value to the journal's readers and the larger scientific community. I recommend the acceptance of publication in the present form to the editors. I wish the authors continued progress and good luck in their scientific endeavors!
